# Discussing the Subjective Well-Being of Hospital Volunteers from a Mental Health Perspective with Health Care System Sustainability during the COVID-19 Pandemic

Kuan-Chieh Tseng [1], Chun-Hao Yen [2], Chin-Shyang Shyu [3], Chih-Hung Tseng [4], Cheng-Ping Li [5,*] and Fang-Wei Lin [6,*]

[1] Master Program in Social Enterprise & Cultural and Creative Industries, College of Humanities and Social Sciences, Providence University, Taichung 43301, Taiwan; jackt72@pu.edu.tw
[2] Department of Leisure Industry Management, National Chin-Yi University of Technology, Taichung 41170, Taiwan; tn332526@hotmail.com
[3] Department of Recreation and Holistic Wellness, MingDao University, Changhua 52345, Taiwan; shaw@mdu.edu.tw
[4] Department of Healthcare Industry Technology Development and Management, National Chin-Yi University of Technology, Taichung 41170, Taiwan; boy217010@hotmail.com
[5] Department of Sports Management, Minghsin University of Science and Technology, Hsinchu 30401, Taiwan
[6] Department of Physical Education Adjunct Instructor, National Taitung University, Taitung 95092, Taiwan
* Correspondence: sm2015@must.edu.tw (C.-P.L.); c757162@gmail.com (F.-W.L.)

**Abstract:** This study explored the subjective well-being of hospital volunteers during the COVID-19 pandemic from a mental health perspective using a health care system sustainability perspective, and adopted the purposive sampling method to conduct a questionnaire survey. A total of 520 questionnaires were distributed, and 500 questionnaires were recovered, with a recovery rate of 96.1%. Of the recovered questionnaires, 43 invalid questionnaires were eliminated, and 457 valid questionnaires were collected, for a valid recovery rate of 91.4%. The data analysis method explored the variable path analysis through descriptive analysis and structural equation modeling. In addition, new data analysis methods have been added to explore the variable path analysis, such as descriptive analysis, discriminant validity, mediation effects, and structural equation modeling. The results show that: (1) the work resources of the hospital volunteers exerted a significant impact on their subjective well-being; (2) the work resources of the hospital volunteers exerted no significant impact on their flow experience; (3) the work resources of the hospital volunteers exerted a significant impact on their leisure involvement; (4) the leisure involvement of the hospital volunteers exerted no significant impact on their subjective well-being; (5) the leisure involvement of the hospital volunteers exerted a significant impact on their flow experience; (6) the flow experience of the hospital volunteers exerted no significant impact on their subjective well-being; and (7) the leisure involvement of the hospital volunteers exerted a mediating effect between work resources and flow experience.

**Keywords:** health care system; sustainability; mental health; leisure involvement; flow experience; subjective well-being

## 1. Introduction

Since the World Health Organization confirmed COVID-19 as a global pandemic, countries around the world have gradually made various efforts to inhibit its infection or co-exist with the virus. Nurses are critical members of the entire medical system, and they play a pivotal role in providing medical care [1]. However, with the COVID-19 epidemic, the lack of medical manpower has forced hospital volunteers to play an important role. If hospitals can effectively take advantage of volunteer manpower, the quality of medical services be ameliorated and the social image of volunteer services can be better shaped [2]. However, through offering voluntary services during this time, their physical,

psychological, and work goals have changed. Liu and Hsieh [3] pointed out impacts of COVID-19 and its related epidemic prevention measures on the management of hospital volunteer services. In other words, with COVID-19, the psychological and physical costs of hospital volunteers' services have grown. This study investigated the influence of changes in work resources on aspects of hospital volunteers' services (such as flow experience and subjective well-being). Furthermore, hospital volunteers carry out helping behaviors out of social public welfare responsibilities. In addition to the voluntary services in their career planning, hospital volunteers have access to leisure activities and obtain a work–life balance. Stebbins [4] divided public leisure into casual leisure and serious leisure. Hospital volunteers engage in voluntary services in hospitals not only for time-killing purposes but also for serious leisure. In other words, they are active in participation, and they treat voluntary services as a leisure activity while being seriously involved. They regard voluntary services as leisure activities and seriously participate in them, so as to obtain a sense of happiness [5] and flow experience [6].

The impact of volunteers' flow experience on subjective well-being has also been demonstrated by relevant research [7]. Therefore, with COVID-19, the impact of changes in hospital volunteers' work resources on their leisure involvement is likely to be one of the key factors for the commitment of voluntary manpower in the future. Furthermore, the influence of hospital volunteers' leisure involvement on flow experience and well-being may also have a bearing on their willingness to persist in voluntary services in the future.

It can be seen from the relevant literature that the research topics on volunteers during the COVID-19 epidemic are diversified. For example, [8] probed into the difficulties faced by Australian volunteers at the beginning of the COVID-19 epidemic. They put forward that the economic recession and growing social isolation appearing in society may increase the demand for certain forms of volunteer services. Tejativaddhana et al. [9] took Thai volunteers as research subjects to further explore the significant contributions of volunteers to disease control. Pickell and Williams [10] probed into the efficacy of virtual voluntary services during the COVID-19 epidemic, and manifested that such services can ease the pressure on medical personnel, reduce the risk of viral infection, and make patients and family life return to normal. Based on the above, during the COVID-19 epidemic, topics about the psychological and physiological costs of hospital volunteers in the workplace and how voluntary services affect their mental health have been rarely discussed. Therefore, this research took hospital volunteers as the research subjects and conducted an empirical study on the subjective well-being of hospital volunteers during the COVID-19 pandemic from a mental health perspective.

People are often confronted with different work resources in the workplace, including physical, psychological, or social resources. Demerouti et al. [11] first proposed the job demands–resources (JD-R) model. Its basic assumption is that each occupation has two types of work characteristics—work demand and work resources, which constitute the overall model applicable to all occupational environments. The theory of work resources originates from the conservation of resources theory, which insists that people will have the motivation to acquire, preserve, and hold resources they consider valuable or the means they should own [12]. Demerouti and Bakker [13] further pointed out that work resources can be divided into the organizational level, interpersonal level, position level, and task level. Specifically, the organizational level includes salary, career opportunities, and job security; the interpersonal level is about supervisors' and colleagues' support and team atmosphere; the job level involves role clarity and the degree of participation in decision making; the task level contains skill diversity, job integrity, task importance, autonomy, and performance feedback. Different levels of job requirements and work resources interact with each other and produce varying effects on organizational members, which can lead to job burnout, work pressure, or work engagement. Under the COVID-19 epidemic, it is worth further exploring whether changes in the work resources of hospital volunteers will have an impact on the completion of tasks.

When people participate in a specific leisure activity and commit their time and energy to it, the psychological state of triggering motivation can be regarded as the degree of involvement in the leisure activity. Havitz and Dimanche [14] demonstrated that leisure involvement is the degree of awareness and cognitive perception between the individual and the activity itself, the tourism destination, or the equipment being used. In other words, when individuals participate in preferred activities, their engagement and concentration can be generated through the cognitive levels of the importance of the activity, the pleasure value, and the symbolic value. From this definition, it can be seen that the deeper people's involvement in leisure is, the higher the degree of concern to be involved in leisure. Hospital volunteers perform voluntary services as a kind of serious leisure, which can represent the degree of involvement of volunteers in unpaid services. Relevant research has also confirmed the impact of the serious leisure of volunteer participants on leisure involvement. For example, Wei [15] took volunteers at the 2018 Taichung World Flora Exposition as the research subjects, and the results show that the serious leisure of volunteers positively and significantly affects leisure involvement. Chen [16] suggested that the mutual progress and happiness of volunteers' serious leisure can remarkably predict leisure involvement. As for the dimension of leisure involvement, most research uses the three dimensions proposed by [17] as the benchmark, namely, attraction, self-expression, and centrality. Attraction refers to the importance and pleasure of a certain activity for participants; self-expression refers to the need of participants to pursue self-realization and the expression of personal identity after participation in the activity; and centrality refers to the subjective identification value of participants' involvement in leisure activities by their lifestyle. Hsu and Chen [18] put forward similar views, indicating that the reasons for different levels of involvement include personal values, emotions, and interests. In the process of engaging in leisure activities, individuals can experience physical and psychological improvement and the enhancement of interpersonal relationships. These individuals' feelings are leisure benefits.

Flow experience is a pleasant and positive psychological state. When individuals are fully engaged in an activity, they will have all kinds of feelings when their skills and external challenges reach a balance [19]. In other words, with a change in the relationship between challenges and skills, individuals' feelings will also vary. On one hand, when the difficulty of the challenges increases, individuals must learn new skills. On the other hand, when personal skills are improved, people will seek more difficult challenges to obtain a flow experience and feel a sense of accomplishment and happiness. While this experience model forms a virtuous cycle, people can obtain leisure experience from flow experience. Csikszentmihalyi [20] pointed out the nine characteristics of flow experience: clear goals, unambiguous feedback, challenge–skill balance, action–awareness merging, concentration on the task at hand, sense of control, loss of self-consciousness, the transformation of time, and automatic experience. Hence, it can be known that the flow experience is not easy to obtain, as it can only be achieved when relevant conditions are ripe. Moneta and Csikszentmihalyi [21] summed up the characteristics of flow experience to illustrate this kind of special leisure experience. First, the function generated from the pleasurable experience depends on subjective experience variables, including challenges and skills. Second, the experience function with a speculative element has no end or extreme, because the flow theory does not limit the optimal experience. Through the flow experience, individuals can continue to pursue harder challenges and greater pleasure. In terms of the measurement of flow experience, [20] first proposed the original flow model and pointed out that skills and challenges are the keys to affect the flow model. When personal skills and challenges are balanced, a flow experience will occur; when skills outweigh challenges, boredom will be generated and interest will be lost; when challenges outweigh skills, anxiety will be produced and control over the situation will be lacking. Therefore, only when the levels of skills and challenges are high and reach a balance will an individual have the flow experience, show interest in it, and pursue a higher level of flow experience. However, the original flow model was designed for more complex sports, and it cannot be

applied to people engaged in general leisure activities. Therefore, Massimini and Carli [22] proposed an eight-dimensional flow model (indifference, concern, anxiety, awakening, fluency, control, boredom, and relaxation) that divides skills and challenges into three levels (low, medium, and high).

In addition to pursuing physical health, people also spare no effort to pursue mental health, which is one of the reasons why well-being is valued. In 1984, Diener [23] suggested that subjective well-being that refers to a state of happiness means having more positive emotions, fewer negative emotions, and a higher sense of life satisfaction. Specifically, having more positive emotions and fewer negative emotions reflects the emotional level of subjective well-being, while having a higher sense of life satisfaction reflects the cognitive level of subjective well-being. Seligman and Csikszentmihalyi [24] opined that subjective well-being is regarded as a valuable experience as well as a positive personal psychological state. Edwards et al. [25] held a similar view, pointing out that subjective well-being is the core indicator of mental health, and is based on the discussion of hedonic well-being. They emphasized that happiness is more than just the absence of mental illness. Happiness is the most ideal manifestation of the human psychological function and experience. In other words, whether individuals feel happy or not can be referred to as a subjective statement of their current physical and psychological state and whether they are satisfied with life. Yen and Xu [26] considered that subjective well-being can be divided into three levels after integrating relevant research results: (1) life satisfaction, which refers to the subjective satisfaction of individuals with their current overall life, including physical and spiritual aspects; (2) positive emotion, which refers to the happy state that individuals experience mentally or emotionally and is often related to social activities, a sense of satisfaction, and pleasant events; and (3) negative emotion, which refers to the unpleasant state that individuals experience mentally or emotionally and is often related to subjective pressure, poor coping styles, health problems, and unpleasant events. From the perspective of these three levels, subjective well-being is a holistic concept that enables people to make statements about their own emotions and further understand their psychological and emotional states.

In terms of the relationships among the variables in this study, Hsieh [27] took pharmaceutical industry workers as the research subjects to explore the relationships among the resources, abilities, and well-being of pharmaceutical industry employees. His study showed that work resources affect subjective well-being. Therefore, the current study assumed that the work resources of hospital volunteers have a significant impact on subjective well-being. Schaufeli and Bakker [28] conducted research on work needs, work resources, and exhaustion, and their research results manifested that work resources affect the flow experience. Based on their research results, the current study assumed that the work resources of hospital volunteers have a significant impact on the flow experience. Lien [29] took educational administrators working in colleges and universities as the research subjects to explore the issues related to leisure involvement. She found that work resources have a significant positive relationship with leisure involvement. Based on the discussion of relevant research, the current study assumed that the work resources of hospital volunteers have a significant impact on leisure involvement. Cheng and Lu [30] took 322 surfing participants as the research subjects in their study, and the research results indicated that the leisure involvement of surfing participants has a positive impact on their subjective well-being. Thus, the current study assumed that the leisure involvement of hospital volunteers has a significant impact on subjective well-being. Tao et al. [31] conducted an empirical analysis of leisure participants in urban parks, and their research results show that leisure involvement has a significant impact on flow experience. According to that research conclusion, the present study assumed that hospital volunteers' leisure involvement has a significant impact on flow experience. Tse et al. [32] revealed that flow experience has a significant impact on subjective well-being in the research targeting a sense of happiness. Therefore, the present study assumed that the flow experience of hospital volunteers has a significant impact on subjective well-being. In addition, the

mediation effect of leisure involvement on work resources and flow experience has also been demonstrated by relevant research [29].

## 2. Research Method

### 2.1. Research Structure

This paper has systematically organized the relationships among variables in the literature review. In examining the interplay of variables in the study of [27,29,33]. The research framework depicted in Figure 1 is constructed based on past relevant studies and the pertinent literature, illustrating the interconnected relationships among variables. The research framework was constructed based on the literature, as shown in Figure 1.

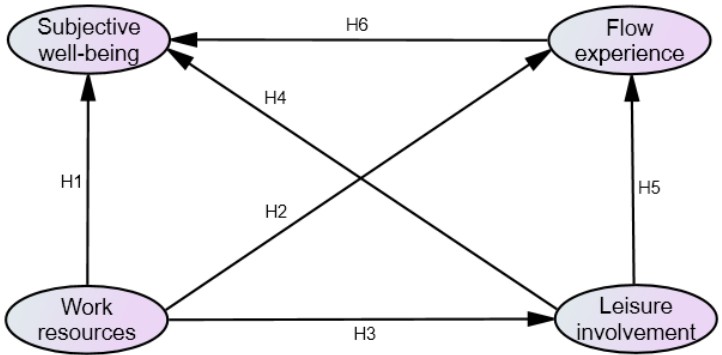

**Figure 1.** Research framework.

### 2.2. Research Hypotheses

The study, based on the literature, proposed a research framework. Subsequently, hospital volunteers, chosen as the research subjects, responded to measurements of work resources, leisure involvement, flow experience, and subjective well-being in the study. The path model within the structural equation modeling framework was then employed to analyze the causal relationship between leisure involvement and subjective well-being among hospital volunteers, validating the research hypothesis. According to the literature discussion and research framework, the following assumptions were proposed:

**H1:** *Work resources have a significant impact on subjective well-being.*

**H2:** *Work resources have a significant impact on flow experience.*

**H3:** *Work resources have a significant impact on leisure involvement.*

**H4:** *Leisure involvement has a significant impact on subjective well-being.*

**H5:** *Leisure involvement has a significant impact on flow experience.*

**H6:** *Flow experience has a significant impact on subjective well-being.*

**H7:** *Leisure involvement has a mediating effect between work resources and flow experience.*

### 2.3. Research Subjects

This study explored the subjective well-being of hospital volunteers during the COVID-19 pandemic from a mental health perspective. This study took hospital volunteers as the research subjects and adopted a questionnaire survey method to collect data. The survey time was from 1 to 31 August 2022. The purposive sampling method was applied, and a total of 520 questionnaires were sent out and 500 questionnaires were recovered. After

eliminating invalid responses, a total of 457 valid questionnaires were collected, for a valid recovery rate of 91.4%.

### 2.4. Research Tools

The content of the questionnaire in this study was revised and based on the relevant literature of [27,29,33]. The questionnaire contained 64 items and was divided into five parts, including personal basic information, work resources, leisure involvement, flow experience, and subjective well-being. This study utilized a five-point Likert scale for scoring, in which each item was given 1 to 5 points to indicate answers ranging from strongly disagree to strongly agree, respectively, as shown in Table A1.

### 2.5. Data Processing and Analysis

In this study, the valid questionnaires were collected and archived. And employed descriptive statistics using SPSS 20.0 statistical software to analyze demographic variables. Additionally, confirmatory factor analysis was conducted using AMOS 20.0 statistical software to validate the measurement model. Furthermore, a path model within the structural equation modeling framework was employed to analyze the causal relationships among variables.

## 3. Research Results

### 3.1. Sample Characteristics

According to Table A2, 457 subjects were concerned in this study. Most of the subjects were women, aged 31–40 years old, were unmarried, had received a college degree, earned NTD 25,001–40,000 monthly, and had been engaged in leisure activities for one to three years.

### 3.2. Analysis of the Measurement Model

The reliability and validity of the questionnaire in this study were tested and analyzed using confirmatory factor analysis (CFA), and the items were revised in line with the questionnaires. With reference to the modification indicators (MI) suggested by [34], observed variables of resources 3–6 and 12 in work resources; observed variables of involvements 2, 3, 5, 7, and 8 in leisure involvement; observed variables of flows 1, 3, 5, 7, 11–16, and 24 in flow experience; and observed variables of well-beings 1 and 5 in subjective well-being were all deleted.

#### 3.2.1. Verification of the Convergence Validity

This study tested the convergence validity of the dimensions including work resources. The factor loads of all dimensions were between 0.66 and 0.94; the composite reliability (CR) values were between 0.71 and 0.94; and the average variance extracted (AVE) values were between 0.50 and 0.83. All of these values conformed to the proposed convergence validity criteria [35–37], indicating this study had good convergence validity, as shown in Tables 1–4.

#### 3.2.2. Differential Validity

This study calculated the confidence interval of the correlation coefficient between the dimensions through the bootstrap method. A value less than 1 would indicate a complete correlation, manifesting a differential validity between dimensions [38]. The results are shown in Appendix A Tables A3–A6.

**Table 1.** Confirmatory factor analysis—work resources.

| Potential Variables | Observed Variables | Non-Standardized Factor Load | Standard Error | C.R | *p* | Factor Load | SMC | Composition Reliability | Average Variation Extraction |
|---|---|---|---|---|---|---|---|---|---|
| | | **Estimate of Model Parameter** | | | | | | **Convergence Validity** | |
| Development of career | Resource 1 | 1.00 | | | | 0.72 | 0.51 | 0.71 | 0.55 |
| | Resource 2 | 1.05 | 0.08 | 13.70 | *** | 0.77 | 0.60 | | |
| Work autonomy | Resource 7 | 1.00 | | | *** | 0.84 | 0.71 | 0.93 | 0.68 |
| | Resource 8 | 0.95 | 0.04 | 21.39 | *** | 0.82 | 0.67 | | |
| | Resource 9 | 0.96 | 0.04 | 22.29 | *** | 0.84 | 0.70 | | |
| | Resource 10 | 0.95 | 0.05 | 20.08 | *** | 0.78 | 0.62 | | |
| | Resource 11 | 1.01 | 0.04 | 22.57 | *** | 0.85 | 0.72 | | |
| | Resource 13 | 1.04 | 0.05 | 22.54 | *** | 0.85 | 0.72 | | |

*** $p < 0.001$. Source: Compiled by this study.

**Table 2.** Confirmatory factor analysis—leisure involvement.

| Potential Variables | Observed Variables | Non-Standardized Factor load | Standard Error | C.R | *p* | Factor Load | SMC | Composition Reliability | Average Variation Extraction |
|---|---|---|---|---|---|---|---|---|---|
| | | **Estimate of Model Parameter** | | | | | | **Convergence Validity** | |
| Attraction of activity | Involvement 1 | 1.00 | | | | 0.74 | 0.55 | 0.75 | 0.50 |
| | Involvement 4 | 0.86 | 0.07 | 12.14 | *** | 0.69 | 0.47 | | |
| | Involvement 6 | 0.92 | 0.08 | 11.77 | *** | 0.71 | 0.50 | | |
| Self-expression and life focus | Involvement 9 | 1.00 | | | | 0.93 | 0.87 | 0.91 | 0.78 |
| | Involvement 10 | 0.80 | 0.03 | 23.12 | *** | 0.79 | 0.63 | | |
| | Involvement 11 | 1.09 | 0.03 | 31.54 | *** | 0.93 | 0.86 | | |

*** $p < 0.001$. Source: Compiled by this study.

**Table 3.** Confirmatory factor analysis—flow experience.

| Potential Variables | Estimate of Model Parameter | | | | | | | Convergence Validity | |
|---|---|---|---|---|---|---|---|---|---|
| | Observed Variables | Non-Standardized Factor Load | Standard Error | C.R | *p* | Factor Load | SMC | Composition Reliability | Average Variation Extraction |
| Targeted engagement | Flow 2 | 1.00 | | | | 0.74 | 0.54 | 0.87 | 0.53 |
| | Flow 4 | 1.11 | 0.07 | 15.30 | *** | 0.74 | 0.55 | | |
| | Flow 6 | 1.00 | 0.07 | 15.06 | *** | 0.73 | 0.54 | | |
| | Flow 8 | 1.13 | 0.08 | 14.64 | *** | 0.71 | 0.50 | | |
| | Flow 9 | 0.94 | 0.07 | 14.39 | *** | 0.71 | 0.50 | | |
| | Flow 10 | 1.04 | 0.07 | 15.04 | *** | 0.74 | 0.55 | | |
| Mastery of skills | Flow 17 | 1.00 | | | | 0.76 | 0.58 | 0.81 | 0.59 |
| | Flow 18 | 1.07 | 0.07 | 15.54 | *** | 0.78 | 0.60 | | |
| | Flow 19 | 1.09 | 0.07 | 15.30 | *** | 0.78 | 0.61 | | |
| Clear action-awareness | Flow 20 | 1.00 | | | | 0.66 | 0.43 | 0.88 | 0.66 |
| | Flow 21 | 1.22 | 0.08 | 15.14 | *** | 0.83 | 0.68 | | |
| | Flow 22 | 1.26 | 0.08 | 15.42 | *** | 0.84 | 0.71 | | |
| | Flow 23 | 1.34 | 0.08 | 16.11 | *** | 0.91 | 0.83 | | |

\*\*\* *p* < 0.001. Source: Compiled by this study.

**Table 4.** Confirmatory factor analysis—well-being.

| Potential Variables | Estimate of Model Parameter | | | | | | | Convergence Validity | |
|---|---|---|---|---|---|---|---|---|---|
| | Observed Variables | Non-Standardized Factor Load | Standard Error | C.R | *p* | Factor Load | SMC | Composition Reliability | Average Variation Extraction |
| Happy emotion | Well-being 2 | 1.00 | | | | 0.94 | 0.88 | 0.93 | 0.83 |
| | Well-being 3 | 0.98 | 0.03 | 35.55 | *** | 0.92 | 0.84 | | |
| | Well-being 4 | 0.97 | 0.03 | 31.66 | *** | 0.88 | 0.78 | | |
| Life satisfaction | Well-being 6 | 1.00 | | | | 0.89 | 0.79 | 0.94 | 0.77 |
| | Well-being 7 | 1.08 | 0.04 | 29.56 | *** | 0.91 | 0.82 | | |
| | Well-being 8 | 1.14 | 0.04 | 27.97 | *** | 0.89 | 0.79 | | |
| | Well-being 9 | 1.02 | 0.04 | 27.68 | *** | 0.88 | 0.78 | | |
| | Well-being 10 | 1.08 | 0.04 | 24.52 | *** | 0.83 | 0.69 | | |

\*\*\* *p* < 0.001. Source: Compiled by this study.

### 3.2.3. Analysis of the Structural Model

This study adopted seven indicators, namely, $\chi^2$ verification, the ratio of $\chi^2$ to the degree of freedom, the goodness of fit index (GFI), the adjusted goodness of fit index (AGFI), the root mean square error of approximation (RMSEA), the comparative fit index (CFI), and the parsimonious goodness-fit-index (PCFI), to test the fit of the overall structural model. From Table 5, it could be seen that the ratio of $\chi^2$ to the degree of freedom was 2.04 through the analysis of the overall structural model in this study, and the smaller this value is, the higher the fit is. The GFI and AGFI values were 0.88 and 0.86, respectively. The RMSEA value was 0.05 (less than 0.08). The CFI and PCFI values were 0.95 and 0.87, respectively, and were within the standard range [35,37,39], indicating the overall structural model of this study embraced a good fit.

**Table 5.** Fit analysis of the overall model.

| Fit Indices | Allowable Range | Research Model | Judgment of Model Fit |
|---|---|---|---|
| $\chi^2$ (Chi-square) | The smaller the better | 1113.74 | |
| Ratio of $\chi^2$ to the degree of freedom | <3 | 2.04 | Qualified |
| GFI | >0.80 | 0.88 | Qualified |
| AGFI | >0.80 | 0.86 | Qualified |
| RMSEA | <0.08 | 0.05 | Qualified |
| CFI | >0.90 | 0.95 | Qualified |
| PCFI | >0.50 | 0.87 | Qualified |

### 3.2.4. Mediating Effect

For the type of mediating effect in this study, specific judgment methods were proposed. If the 95% confidence interval of the indirect effect value did not contain 0, the indirect effect was significant; that is, there was a mediating effect. Moreover, if the 95% confidence interval of the direct effect value contained 0, the direct effect was not significant, indicating a complete mediating effect [40,41]. The results are shown in Table 6.

**Table 6.** Summary of the mediating effects.

| | Estimate | 95% Confidence Interval | | |
|---|---|---|---|---|
| **Indirect Effect** | | **BC/PC *p*-Value** | **BC** | **PC** |
| Work resources->leisure involvement->flow experience | 0.83 | 0.001/0.001 | 0.56–1.36 | 0.56–1.33 |
| Direct effect | | | | |
| Work resources $\rightarrow$ leisure involvement | 0.85 | 0.001/0.001 | 0.77–0.91 | 0.77–0.91 |
| Work resources $\rightarrow$ flow experience | −0.10 | 0.56/0.59 | −0.63–0.21 | −0.62–0.21 |
| Leisure involvement $\rightarrow$ flow experience | 0.98 | 0.001/0.001 | 0.69–1.48 | 0.68–1.47 |
| Total effect | | | | |
| Work resources $\rightarrow$ flow experience | 0.73 | 0.001/0.001 | 0.63–0.82 | 0.63–0.82 |

BC: bias-corrected percentile method. PC: percentile method.

According to Figure 2 and Table 7, H1 was supported; that is, work resources had a significant impact on subjective well-being. This research result is the same as that of [27]. The possible reason is that the hospitals provided the subject volunteers with diverse work content and flexible service time; therefore, they could obtain a fuller life with more positive emotions. Nielsen, Nielsen, Ogbonnaya, Känsälä, Saari and Isaksson [42] also held the same

argument. Their research results found workplace resources at the individual, group, leader, and organizational levels that are related to both employee well-being and organizational performance. Resources at any of the four levels were related to both employee well-being and performance.

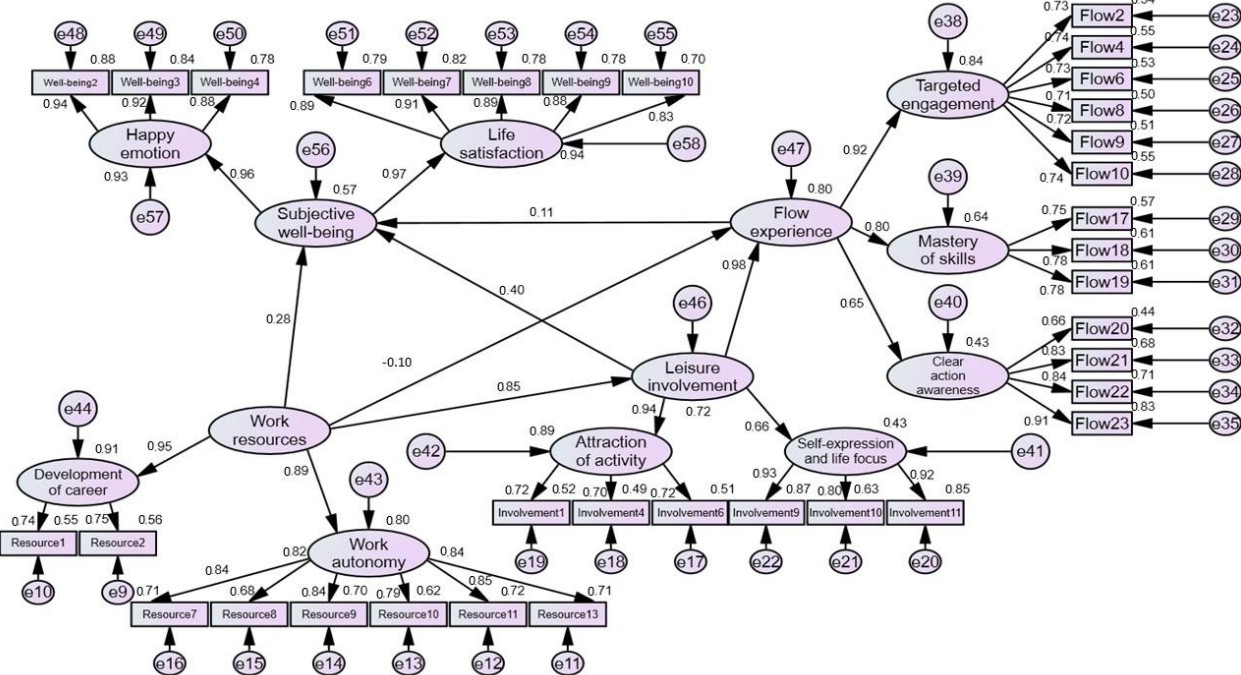

**Figure 2.** Model of the relationships among the work resources, leisure involvement, flow experience, and subjective well-being of hospital volunteers under the COVID-19 epidemic. Annotations: Development of career: Resource 1, Resource 2; Work autonomy: Resource 7, Resource 8, Resource 9, Resource 10, Resource 11, Resource 13; Attraction of activity: Involvement 1, Involvement 4, Involvement 6; Self-expression and life focus: Involvement 9, Involvement 10, Involvement 11; Targeted engagement: Flow 2, Flow 4, Flow 6, Flow 8, Flow 9, Flow 10; Mastery of skills: Flow 17, Flow 18, Flow 19; Clear action-awareness: Flow 20, Flow 21, Flow 22, Flow 23; Happy emotion: Well-being 2, Well-being 3, Well-being 4; Life satisfaction: Well-being 6, Well-being 7, Well-being 8, Well-being 9, Well-being 10.

**Table 7.** Empirical results of the research hypotheses.

| Hypothesis | Path Relationship | Path Value | Supported or Not |
|:---:|---|:---:|:---:|
| 1 | Work resources have a significant impact on subjective well-being. | 0.28 * | Supported |
| 2 | Work resources have a significant impact on flow experience. | −0.10 | Not supported |
| 3 | Work resources have a significant impact on leisure involvement. | 0.85 * | Supported |
| 4 | Leisure involvement has a significant impact on subjective well-being. | 0.40 | Not supported |
| 5 | Leisure involvement has a significant impact on flow experience. | 0.98 * | Supported |
| 6 | Flow experience has a significant impact on subjective well-being. | 0.11 | Not supported |
| 7 | Leisure involvement has a mediating effect between work resources and flow experience. | * | Supported |

* $p < 0.05$.

H2 was not supported; in other words, work resources had a significant impact on flow experience. This research result is different from that of [28]. The possible reason is that under the COVID-19 epidemic, the subject volunteers considered that they could not excel in volunteer services as scheduled due to an increase in their psychological and physiological costs, thus reducing the flow experience. Farina, Rodrigues and Huts [43] also proposed a similar argument that flow and engagement are linked to superior results in business, work performance, and life satisfaction. It follows that work resources play an important role in the physiological and psychological well-being of employees.

H3 was supported, meaning work resources had a significant impact on leisure involvement. This research result is the same as that of [29]. The possible reason is that when the hospitals provided an environment in which the subject volunteers could achieve their work goals easier and reduce their physical and psychological costs associated with work, they were more willing to put effort into voluntary services. Strassburger, Wachholz, Peters, Schnitzer and Blank [44] also mentioned a similar view. Their studies found that organizational leisure benefits can be a multifaceted organizational resource to (1) facilitate employees' leisure participation, (2) boost employees' recovery, or (3) meet the employees' need for workplace fun.

H4 was not supported; in other words, leisure involvement had no significant impact on the flow experience. This research result is different from that of [45]. The possible reason might be that although the subject volunteers were engaged in voluntary services, the work content may have been too simple or monotonous, leading to an imbalance between challenges and abilities and a lack of flow experience. Chang [46] added the factor of gender to explore this topic, and suggested that leisure involvement has a stronger effect on the level of flow experience in male participants, while the factors of "centrality to lifestyle" and "self-expression" have statistically significant influences on the flow experience of female participants. It was worth further exploring whether the establishment of the four hypotheses in this study was influenced by gender.

H5 was supported, indicating that leisure involvement had a significant impact on subjective well-being. This research result is the same as that of [47]. The possible reason is that the subject volunteers were engaged in voluntary services. Under the concept of giving more than receiving, the volunteers were able to obtain more positive emotions. At the same time, because of the enrichment of their lives, the volunteers' life satisfaction was high, and thus higher subjective well-being could be attained. Lin Chen and Kuo [48] also supported the above viewpoint in their research findings, indicating that recreational activities with greater leisure involvement lead to a high level of subjective well-being.

H6 was not supported; in other words, flow experience had no significant impact on subjective well-being. This research result is different from that of [7]. The possible factor is that the service content of the subject volunteers was too monotonous, and the subjects' challenges and abilities were not balanced. Services were provided only to pass the time, resulting in the failure to produce higher subjective well-being. Sahoo [49] used managers from software organizations as samples to conduct research and found a positive association between flow and well-being. It was worth further exploring whether H6 in this study was not supported due to differences in job positions.

H7 was supported, indicating that leisure involvement had a mediating effect on work resources and flow experience. This research result is the same as that of [29]. The possible reason is that when the hospitals provided more support to the subject volunteers (such as epidemic prevention materials, flexible scheduling times, and positive incentives), the volunteers became more willing to participate in different challenges of voluntary services because of sufficient resources. Moreover, due to the high involvement of the hospital volunteers, their flow experience could be further affected. This research result also echoed the arguments of Farina, Rodrigues, and Huts [43] that leisure involvement plays an important role in a company or organization.

## 4. Discussion

### 4.1. Conclusions

This study took hospital volunteers as the research subjects and conducted an empirical study on the subjective well-being of hospital volunteers during the COVID-19 pandemic from a mental health perspective. From the research results, it was concluded that hospital volunteers' work resources have a significant impact on subjective well-being and leisure involvement. The leisure involvement of hospital volunteers has a significant impact on flow experience and a mediating effect between work resources and flow experience. Work resources have no significant impact on flow experience, and leisure involvement and flow experience have no significant impact on subjective well-being. The results show that the research framework constructed according to the theory had some corresponding results after the empirical research.

### 4.2. Suggestions

#### 4.2.1. Suggestions for Hospital Volunteers

The results of this study demonstrate that leisure involvement exerts a significant impact on flow experience. Therefore, it is suggested that hospital volunteers continue to engage in training courses to improve themselves. Especially with the progress of digital technology, more medical-related services are becoming electronic and paperless, such as Taiwan's National Health Insurance app, the E-health Pay app, and the network registration system. Such training courses, in addition to reducing the digital gap, could allow volunteers to better assist patients or their families in completing relevant procedures and processes. For example, under the COVID-19 epidemic, in addition to Taiwan's COVID-19 Vaccination Record Cards, the National Health Insurance Mobile APP also contains relevant content vaccination records. Familiarity with the operation of the National Health Insurance Mobile APP could enable volunteers to provide information to patients or their families in a timely manner when giving voluntary services. In the meantime, when hospitals need relevant volunteer manpower, after being equipped with relevant skills that are balanced with challenges, volunteers could have opportunities to produce a flow experience and show more enthusiasm when providing services to patients or their family members.

The results of this study reveal that flow experience had no significant impact on subjective well-being. Therefore, it is suggested that hospital volunteers not only improve themselves to provide better services and counseling but also seek emotional outlets for matters they have to face in the hospital every day, such as birth, aging, illness, and death, as well as the anxiety of patients and their family members. Such emotional outlets could help volunteers avoid emotional disorders that increase their psychological costs and cause emotional exhaustion. Therefore, it is suggested that hospital volunteers participate in social activities and better integrate into volunteer groups to benefit from having closer emotional connections with hospital volunteer groups. Volunteers are also advised to enhance their positive emotions and mental health by sharing their experiences with other volunteers, which can lead to satisfaction and a richer life.

This study showed that leisure involvement had a mediating effect between work resources and flow experience. It is suggested that hospital volunteers continue to pursue the sense of pleasure and achievement brought by voluntary services based on the appropriate resources provided by the hospitals, such as sharing the service experience of special cases with volunteer groups. When these volunteers share their own experiences with new volunteers, voluntary services in hospitals can be promoted constantly. That is, they offer services to the served and gain self-realization from growth opportunities. Moreover, when hospital volunteers commit more and their professional abilities and task difficulties reach a balance, they will be better able to determine what kind of volunteer activities they want to participate in, and they will realize they have more opportunities to learn new things and make progress.

### 4.2.2. Suggestions for Hospital Administrators

The results of this study indicate that work resources had a significant impact on subjective well-being and leisure involvement. Therefore, it is suggested that hospital management units adjust or modify the service content, management system, and supervision of hospital volunteers in response to COVID-19. These adjustments can avoid poor work design or excessive work requirements that deplete the physical and psychological energies of hospital volunteers and cause them to be unable to obtain happiness from or become unwilling to participate in voluntary activities. Specifically, hospital management units can re-plan their volunteer manpower needs, as well as the service times and places, according to the retention and suspension of stations and the number of people on duty during the epidemic prevention period. For example, volunteer manpower can be arranged at the lobby service desk, the emergency and health inspection center, and other places. In addition, hospitals should also implement epidemic prevention and control measures, urge volunteers to perform disinfection and cleaning before and after providing services as well as wear masks, and require them to comply with epidemic prevention regulations to reduce the risk of infection. Furthermore, hospital administrators can continue to communicate with hospital volunteers. For example, hospitals are often tightly controlled against epidemic infection. In particular, admission staff must check health insurance cards, inquire about travel histories, and measure temperatures. Hospital administrators can also timely convey the idea that by volunteering in hospitals, volunteers can obtain more of the latest information about the COVID-19 epidemic and strengthen their sense of self-protection. In other words, through the support of hospital management units, hospital volunteers can provide appropriate resources (such as epidemic prevention materials), so that they can see that voluntary services are under acceptable control. In this way, hospital volunteers can trigger their positive psychological motivation, reduce their work pressure and emotional burden, and achieve happiness by providing voluntary services while promoting mental health.

### 4.2.3. Limitations and Future Work

This study only provided self-reported questionnaires to volunteers. However, the organizational cultures and systems of hospitals can vary, such as the leadership models and incentive systems used for volunteer groups, which could not be explored in depth by this study. This is one of the research limitations of this study. In addition, the emotional exhaustion faced by volunteers in different hospital units can also vary. For example, patients and their families in the health examination department and those at the lobby service desk have different characteristics. Limited by the length of this paper, this study did not include the above issues in the discussion. The inclusion of management system issues in future research on the psychological aspect of hospital volunteers could help to provide a more complete content. In addition, according to the classification of the Ministry of Health and Welfare, Taiwan's medical institutions are currently divided into four levels: medical centers, regional hospitals, district hospitals, and primary clinics. The work resources provided by different levels of medical institutions are different. Especially under the influence of the COVID-19 pandemic, various medical institutions have gradually adjusted their organizational resources and manpower to meet the needs. Therefore, the results can only reflect the characteristics of the sample of this study and cannot be generalized, which is a limitation of this study. It is suggested that subsequent research can conduct study or case analysis on the volunteer management topics at the same level of medical institutions, so as to accurately reflect the current situation and serve as a reference for medical institutions in managing volunteers in the future.

**Author Contributions:** Conceptualization, F.-W.L. and K.-C.T.; Methodology, F.-W.L., C.-H.T. and C.-H.Y.; Formal analysis, C.-H.T., C.-H.Y. and C.-S.S.; Investigation, C.-H.Y. and C.-S.S.; Data curation, C.-S.S. and C.-H.Y.; Supervision, C.-P.L.; Writing—original draft, K.-C.T.; Writing—review & editing, C.-P.L. All authors have read and agreed to the published version of the manuscript.

**Funding:** This research received no external funding.

**Institutional Review Board Statement:** Not applicable.

**Informed Consent Statement:** Informed consent was obtained from all subjects involved in the study.

**Data Availability Statement:** The data will not be made public, and its use and access rights will be managed by the corresponding author.

**Conflicts of Interest:** The authors declare no conflict of interest.

### Appendix A

**Table A1.** Questionnaire on discussing the subjective well-being of hospital volunteers during the COVID-19 pandemic from a mental health perspective.

| Dimension | Observed Variables | Contents |
|---|---|---|
| Flow experience | Flow1 | 1. I have a clear goal of participating in hospital volunteer activities. |
| | Flow2 | 2. I am clear about what I want to accomplish when I participate in hospital volunteer activities. |
| | Flow3 | 3. I feel that participating in hospital volunteer activities is a worthwhile experience. |
| | Flow4 | 4. I know what I want to do when I participate in hospital volunteer activities. |
| | Flow5 | 5. I have a deep feeling about the experience of volunteering at the hospital. |
| | Flow6 | 6. I would like to repeat the experience of volunteering at the hospital. |
| | Flow7 | 7. I found the experience of participating in hospital volunteer activities enjoyable. |
| | Flow8 | 8. I have a strong desire to participate in hospital volunteer activities. |
| | Flow9 | 9. When I participate in hospital volunteer activities, my attention is focused on the activities. |
| | Flow10 | 10. I have the ability to face challenges when volunteering in a hospital. |
| | Flow11 | 11. I have the ability to perform the hospital volunteer activities that I am currently engaged in. |
| | Flow12 | 12. When I participate in hospital volunteer activities, I feel that I am able to take on more difficult activities than I am doing now. |
| | Flow13 | 13. I feel that my volunteer skills are getting better and better as I participate in hospital volunteer activities. |
| | Flow14 | 14. I can fully demonstrate my abilities when participating in hospital volunteer activities. |
| | Flow15 | 15. I feel in control when I am involved in hospital volunteer activities. |
| | Flow16 | 16. I am in control of what I am responsible for when I participate in hospital volunteer activities. |
| | Flow17 | 17. I do not worry about not doing well enough when I participate in hospital volunteer activities. |
| | Flow18 | 18. I can remember everything that happens when I am involved in hospital volunteering. |
| | Flow19 | 19. When I am involved in hospital volunteering, I do not have to deal with my current problems in life. |
| | Flow20 | 20. Time passes without me realizing it when I am involved in hospital volunteering. |
| | Flow21 | 21. My involvement in hospital volunteer activities is spontaneous. |
| | Flow22 | 22. I volunteer in hospitals naturally and do not need to force myself. |
| | Flow23 | 23. I know how to do things well when I am involved in hospital volunteer activities. |
| | Flow24 | 24. When I am involved in hospital volunteer activities, I am able to carry out hospital volunteer activities spontaneously. |

**Table A1.** *Cont.*

| Dimension | Observed Variables | Contents |
|---|---|---|
| Leisure involvement | Involvement 1 | 1. Getting involved in hospital volunteer activities is important to me. |
| | Involvement 2 | 2. Getting involved in hospital volunteer activities makes me feel fulfilled. |
| | Involvement 3 | 3. Getting involved in hospital volunteer activities makes me feel happy. |
| | Involvement 4 | 4. I am interested in participating in hospital volunteer activities. |
| | Involvement 5 | 5. It is a pleasure for me to participate in hospital volunteer activities. |
| | Involvement 6 | 6. I can show my true self when I engage in hospital volunteer activities. |
| | Involvement 7 | 7. I can tell what kind of person someone is by the way he or she engages in hospital volunteer activities. |
| | Involvement 8 | 8. Hospital volunteering has become a personal characteristic of mine and has formed an impression of me. |
| | Involvement 9 | 9. I find that hospital volunteer activities are closely related to my life. |
| | Involvement 10 | 10. I find that hospital volunteering plays an important role in my life. |
| | Involvement 11 | 11. I find that most of my life is related to my hospital volunteer activities. |
| Work resources | Resource1 | 1. I have the opportunity to develop my strengths in hospital volunteer activities. |
| | Resource2 | 2. I am able to fully express myself in hospital volunteer activities. |
| | Resource3 | 3. I have the opportunity to learn new things when I volunteer in hospitals. |
| | Resource4 | 4. I can decide by myself how I want to work in hospital volunteer activities. |
| | Resource5 | 5. I can decide by myself the time I want to leave the venue of hospital volunteer activities. |
| | Resource6 | 6. I can decide my own goals for my hospital volunteer activities. |
| | Resource7 | 7. I can decide the order of my hospital volunteer activities. |
| | Resource8 | 8. I can evaluate my own hospital volunteer activities. |
| | Resource9 | 9. I can suspend my hospital volunteer activities. |
| | Resource10 | 10. I can decide the amount of hospital volunteer work I need to accomplish within a certain period of time. |
| | Resource11 | 11. I can decide by myself the pace of my hospital volunteer activities. |
| | Resource12 | 12. I can decide by myself how long I want to do my hospital volunteering activities. |
| | Resource13 | 13. I can decide by myself what kind of hospital volunteering activities I want to participate. |
| Subjective well-being | Well-being1 | 1. I feel that I am living my life in high spirits. |
| | Well-being2 | 2. I feel that I am living a very happy life. |
| | Well-being3 | 3. I feel very relaxed about my life. |
| | Well-being4 | 4. I feel that my life is full of joy. |
| | Well-being5 | 5. I feel confident. |
| | Well-being6 | 6. I am living the lifestyle I want to live. |
| | Well-being7 | 7. I am very satisfied with my current lifestyle. |
| | Well-being8 | 8. I don't want to change my current life. |
| | Well-being9 | 9. I feel in control of my life. |
| | Well-being10 | 10. I enjoy life every day. |

**Table A2.** Subject characteristics.

| Variable | Classification Standard | Number of Samples | Percentage (%) | Cumulative Percentage (%) |
|---|---|---|---|---|
| Gender | Male | 219 | 47.9 | 47.9 |
| | Female | 238 | 52.1 | 100.0 |
| Age | 20 (including) to 30 years old | 75 | 16.4 | 16.4 |
| | 31–40 years old | 162 | 35.4 | 51.9 |
| | 41–50 years old | 87 | 19.0 | 70.9 |
| | 50–60 years old | 101 | 22.1 | 93.0 |
| | 61 years old and above | 32 | 7.0 | 100.0 |
| Educational level | Junior high school and below | 65 | 14.2 | 14.2 |
| | Senior and vocational high school | 157 | 34.4 | 48.6 |
| | College and junior college | 202 | 44.2 | 92.8 |
| | Master and Doctor | 33 | 7.2 | 100.0 |
| Marital status | Married | 153 | 33.5 | 33.5 |
| | Unmarried | 304 | 66.5 | 100.0 |
| Monthly income | NTD 25,000 and below | 99 | 21.7 | 21.7 |
| | NTD 25,001–40,000 | 157 | 34.4 | 56.0 |
| | NTD 40,001–55,000 | 125 | 27.4 | 83.4 |
| | NTD 55,001–70,000 | 48 | 10.5 | 93.9 |
| | NTD 70,001–85,000 | 15 | 3.3 | 97.2 |
| | NTD 85,001 and above | 13 | 2.8 | 100.0 |
| Number of years participating in leisure activities | 1–3 years | 148 | 32.4 | 32.4 |
| | 4–6 years | 143 | 31.3 | 63.7 |
| | 7–9 years | 80 | 17.5 | 81.2 |
| | 10–12 years | 52 | 11.4 | 92.6 |
| | 13–15 years | 23 | 5.0 | 97.6 |
| | 16 years and above | 11 | 2.4 | 100.0 |

**Table A3.** 95% trust interval table of bootstrap correlation coefficient—work resources.

| | | | Bias-Corrected | | | Percentile Method | |
|---|---|---|---|---|---|---|---|
| | | | Estimate | Lower Bound | Upper Bound | Lower Bound | Upper Bound |
| Development of career | <--> | Work autonomy | 0.85 | 0.77 | 0.92 | 0.77 | 0.92 |

Source: Compiled by this study.

**Table A4.** 95% trust interval table of bootstrap correlation coefficient—leisure involvement.

| | | | Bias-Corrected | | | Percentile Method | |
|---|---|---|---|---|---|---|---|
| | | | Estimate | Lower Bound | Upper Bound | Lower Bound | Upper Bound |
| Attraction of activity | <--> | Self-expression and life focus | 0.62 | 0.54 | 0.70 | 0.54 | 0.70 |

**Table A5.** 95% trust interval table of bootstrap correlation coefficient—flow experience.

| | | | Bias-Corrected | | | Percentile Method | |
|---|---|---|---|---|---|---|---|
| | | | Estimate | Lower Bound | Upper Bound | Lower Bound | Upper Bound |
| Targeted engagement | <--> | Mastery of skills | 0.76 | 0.68 | 0.83 | 0.68 | 0.83 |
| Targeted engagement | <--> | Clear action-awareness | 0.58 | 0.50 | 0.67 | 0.49 | 0.66 |
| Mastery of skills | <--> | | 0.48 | 0.39 | 0.58 | 0.38 | 0.56 |

Source: Compiled by this study.

**Table A6.** 95% trust interval table of bootstrap correlation coefficient—subjective well-being.

| | | | Bias-Corrected | | | Percentile Method | |
|---|---|---|---|---|---|---|---|
| | | | Estimate | Lower Bound | Upper Bound | Lower Bound | Upper Bound |
| Happy emotion | <--> | Life satisfaction | 0.93 | 0.90 | 0.96 | 0.90 | 0.96 |

Source: Compiled by this study.

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
