# Peer review of "Discussing the Subjective Well-Being of Hospital Volunteers from a Mental Health Perspective with Health Care System Sustainability during the COVID-19 Pandemic"

_sustainability, doi:10.3390/su16062404_

Round 1

Reviewer 1 Report (Previous Reviewer 4)

Comments and Suggestions for Authors

Improved tables and text 

Author Response

Reviewers’ Comments and Suggestions for Author (Round 3)

9 March 2024

Dear Reviewer,

Thank you for the constructive suggestions and comments on our manuscript(ID:sustainability-2878926). The suggestions and comments are helpful for improving the manuscript. We are submitting the revised version of the manuscript with our responses to the suggestions and comments by the reviewer. Many thanks for your guidance.

Our responses to each suggestion and comment are as follows, and they are presented in red texts with a grey background color in the revised manuscript.

Comments and Suggestions for Authors

Improved tables and text

Response:

Thank you very much for your comments and suggestion. The modifications are as follows:

Corrected and improved tables and text

Reviewer 2 Report (Previous Reviewer 3)

Comments and Suggestions for Authors

Thank you. Now the data analysis method has become clear to me.

However, I still believe that the data from one questionnaire is not enough to draw conclusions about cause-and-effect relationships.

Author Response

Reviewers’ Comments and Suggestions for Author (Round 3)

9 March 2024

Dear Reviewer,

Thank you for the constructive suggestions and comments on our manuscript(ID:sustainability-2878926). The suggestions and comments are helpful for improving the manuscript. We are submitting the revised version of the manuscript with our responses to the suggestions and comments by the reviewer. Many thanks for your guidance.

Our responses to each suggestion and comment are as follows, and they are presented in red texts with a grey background color in the revised manuscript.

Thank you. Now the data analysis method has become clear to me.

However, I still believe that the data from one questionnaire is not enough to draw conclusions about cause-and-effect relationships.

Response:

Thank you very much for your comments and suggestion. The modifications are as follows:

New contents have been added.

Reviewer 3 Report (New Reviewer)

Comments and Suggestions for Authors

The subjective well-being of hospital volunteers during the COVID 19 pandemic was analyzed. The article presents interesting and important data. However, several issues should be analyzed.

 Abstract:

1.       In the abstract, instead of the information "The research hypotheses of this study were 25 tested through SPSS 20.0 and AMOS 20.0 statistical analysis software", information on statistical methods should be provided.

Introduction:

1.       The introduction should be significantly shortened. The purpose of the research was well presented.

Research Method:

1.       The abstract states "A total of 520 questionnaires were distributed, and 500 questionnaires were recovered, with a recovery rate of 96.1%."

2.       However, in the text (line 258) it was entered "The purposive sampling method was 258 applied, and a total of 500 questionnaires were sent out. After eliminating invalid responses, a total of 457 valid questionnaires were collected, for a valid recovery rate of 260 91.4%.” This information needs to be clarified.

Discussion:

1.       Table 7 should be presented in the Results section

2.       Figure 2 should be presented in the Results section and annotated appropriately because the markings are not entirely clear

3.       The results were insufficiently discussed compared to the studies of other authors.

4.       Limitations of the work should be highlighted.

General thoughts:

1.       Tables:

a.       The tables are unreadable

b.       Please check the order of the tables in the text.

c.       Journal allows font size reduction (but no less than 8 pt. in size)

d.       In some tables you can add horizontal lines

e.       Data in the tables should be better arranged, e.g. "Nonstandardized load factor" (Table 3)

f.        The numbering of Tables A3-A6 described in the paragraph on lines 336-339 is unclear

g.       There are no appropriate table titles in subsection 3.2.2.

2.       Font unification (selected text, different fonts and font sizes appear, and tables appear in larger integers)

3.       Line 286, 352 – incomplete subchapter number

4.       The subsection should be written in italics, in accordance with the previous template: “Verification of the convergence validity” or “Suggestions for hospital volunteers”

5.       Authors' individual contributions must be provided

6.       The record of references should be corrected in accordance with the journal's requirements

Author Response

Reviewers’ Comments and Suggestions for Author (Round 3)

9 March 2024

Dear Reviewer,

Thank you for the constructive suggestions and comments on our manuscript(ID:sustainability-2878926). The suggestions and comments are helpful for improving the manuscript. We are submitting the revised version of the manuscript with our responses to the suggestions and comments by the reviewer. Many thanks for your guidance.

Our responses to each suggestion and comment are as follows, and they are presented in red texts with a grey background color in the revised manuscript.

Comments and Suggestions for Authors

The subjective well-being of hospital volunteers during the COVID 19 pandemic was analyzed. The article presents interesting and important data. However, several issues should be analyzed.

Response:

Thank you very much for your comments and suggestion. The modifications are as follows:

 Abstract:

  1. In the abstract, instead of the information “The research hypotheses of this study were 25 tested through SPSS 20.0 and AMOS 20.0 statistical analysis software”, information on statistical methods should be provided.

Line28-29

The following content has been deleted:

The research hypotheses of this study were tested through SPSS 20.0 and AMOS 20.0 statistical analysis software.

The following content has been added to the abstract:

Line31-33

In addition, new data analysis methods have been added to explore the variable path analysis, such as descriptive analysis, discriminant validity, mediation effects, and structural equation modeling.

Introduction:

  1. The introduction should be significantly shortened. The purpose of the research was well presented.

Some paragraphs in the introduction have been deleted and marked with a strikethrough.

Line48-52

Although the spread of COVID-19 has slowed down, Taiwan’s patients with moderate and severe COVID-19, babies who are less than three months old and have a fever, as well as confirmed cases who have been hospitalized due to other diseases, still need to be hospitalized for treatment. Therefore, the load on medical sources has been a focus of social attention.

Line57-60

At present, hospitals around the world have developed management models for volunteer recruitment, education, and training, as well as service divisions. Apart from raising the flexibility of manpower utilization, volunteers have also made a great difference in hospitals under the COVID-19 epidemic.

Line62-66

Liu and Hsieh [3] pointed out that the impacts of COVID-19 and its related epidemic prevention measures on the management of hospital volunteer services are as follows: (1) the risk of infection is increased; (2) inconvenience is caused by epidemic prevention and control measures; (3) COVID-19 is unpredictable and recurring; and (4) psychological pressure is created due to stigma about the disease.

Line73-79

As for the latter, amateurs, hobbyists, and volunteers systematically engage in certain activities, devote career-like attitudes to their activities, and take opportunities to acquire and display special skills, knowledge, and experience. They would even search systematically and intentionally for opportunities to participate in this kind of activities. Such activities are valuable, interesting, and voluntary for participants who are allowed to acquire and display special skills and knowledge in their career experiences. In other words,

Research Method:

  1. The abstract states “A total of 520 questionnaires were distributed, and 500 questionnaires were recovered, with a recovery rate of 96.1%.”

The following content has been added to the abstract:

Line27-28

Of the recovered questionnaires, 43 invalid questionnaires were eliminated, and 457 valid questionnaires were collected, for a valid recovery rate of 91.4%.

  1. However, in the text (line 258) it was entered “The purposive sampling method was 258 applied, and a total of 500 questionnaires were sent out. After eliminating invalid responses, a total of 457 valid questionnaires were collected, for a valid recovery rate of 260 91.4%.” This information needs to be clarified.

The content has been revised to:

Line266-268

The purposive sampling method was applied, and a total of 520 questionnaires were sent out and 500 questionnaires were recovered.

Discussion:

  1. Table 7 should be presented in the Results section

This part has been revised to Line389

  1. Figure 2 should be presented in the Results section and annotated appropriately because the markings are not entirely clear

The cells in Figure 2 have been enlarged, and additional annotations have been added as follows:

Line375-384

Annotations:

Development of career: Resource 1, Resource 2

Work autonomy: Resource 7, Resource 8, Resource 9, Resource 10, Resource 11, Resource 13

Attraction of activity: Involvement 1, Involvement 4, Involvement 6

Self-expression and life focus: Involvement 9, Involvement 10, Involvement 11

Targeted engagement: Flow 2, Flow 4, Flow 6, Flow 8, Flow 9, Flow 10

Mastery of skills: Flow 17, Flow 18, Flow 19

Clear action-awareness: Flow 20, Flow 21, Flow 22, Flow 23

Happy emotion: Well-being 2, Well-being 3, Well-being 4

Life satisfaction: Well-being 6, Well-being 7, Well-being 8, Well-being 9, Well-being 10

  1. The results were insufficiently discussed compared to the studies of other authors.

The contents have been supplemented and are marked in red.

Line 396-400

Nielsen, Nielsen, Ogbonnaya, Känsälä, Saari & Isaksson [42] also held the same argument. Their research results found workplace resources at the individual, group, leader, and organizational levels that are related to both employee well-being and organizational performance. Resources at any of the four levels were related to both employee well-being and performance.

Line405-408

Farina, Rodrigues & Huts [43] also proposed a similar argument that flow and engagement are linked to superior results in business, work performance and life satisfaction. It follows that work resources play an important role in the physiological and psychological well-being of employees.

Line413-417

Strassburger, Wachholz, Peters, Schnitzer & Blank [44] also mentioned a similar view. Their studies found that organizational leisure benefits can be a multifaceted organizational resource to (1) facilitate employees’ leisure participation, (2) boost employees’ recovery or (3) meet the employees’ need for workplace fun.

Line422-427

Chang [46] added the factor of gender to explore this topic, and suggested that leisure involvement has a stronger effect on the level of flow experience in male participants, while the factors of “centrality to lifestyle” and “self-expression” have statistically significant influences on the flow experience of female participants. It was worth further exploring whether the establishment of the four hypotheses in this study was influenced by gender.

Line433-435

Lin Chen & Kuo [48] also supported the above viewpoint in their research findings, indicating that recreational activities with greater leisure involvement lead to a high level of subjective well-being.

Line440-443

Sahoo [49] used managers from software organizations as samples to conduct research and found a positive association between flow and well-being. It was worth further exploring whether H6 in this study was not supported due to differences in job positions.

Line450-452

This research result also echoed the arguments of Farina, Rodrigues and Huts [43] that leisure involvement plays an important role in a company or organization.

  1. Limitations of the work should be highlighted.’

Line535

The section title Relevant research in the future has been changed to Limitations and Future Work, and new contents have been added.

Line539

This is one of the research limitations of this study. Besides, the emotional exhaustion faced by volunteers in different hospital units can also vary.

Line545-555

In addition, according to the classification of the Ministry of Health and Welfare, Taiwan's medical institutions are currently divided into four levels: medical centers, regional hospitals, district hospitals, and primary clinics. The work resources provided by different levels of medical institutions are different. Especially under the influence of the COVID-19 pandemic, various medical institutions have gradually adjusted their organizational resources and manpower to meet the needs. Therefore, the results can only reflect the characteristics of the sample of this study and cannot be generalized, which is a limitation of this study. It is suggested that subsequent research can conduct study or case analysis on the volunteer management topics at the same level of medical institutions, so as to accurately reflect the current situation and serve as a reference for medical institutions in managing volunteers in the future.

Six references have been added:

Line662-667

Nielsen, K., Nielsen, M. B., Ogbonnaya, C., Känsälä, M., Saari, E., & Isaksson, K. Workplace resources to improve both employee well-being and performance: A systematic review and meta-analysis. Work & Stress, 2017, 31, 101-120.

Farina, L. S. A., Rodrigues, G. D. R., & Hutz, C. S. Flow and engagement at work: A literature review. Psico-USF, 2018, 23, 633-642.

Strassburger, C., Wachholz, F., Peters, M., Schnitzer, M., & Blank, C. Organizational leisure benefits–a resource to facilitate employees’ work-life balance?. Employee Relations: The International Journal, 2023, 45, 585-602.

Line670-671

Chang, H. Gender differences in leisure involvement and flow experiencein professional extreme sport activities. World Leisure J. 2016, 59, 124–139.

Line675-678

Lin, H. C., Chen, K. Y., & Kuo, K. P. Relationship between leisure involvement and subjective well-being: moderating effect of spousal support. South African Journal for Research in Sport, Physical Education and Recreation, 2013, 36, 131-146.

Sahoo, F. M. (2015). Flow experience and workplace well-being. Journal of the Indian Academy of Applied Psychology, 2015, 41, 189.

General thoughts:

  1. Tables:

  1. The tables are unreadable

The tables have been revised.

  1. Please check the order of the tables in the text.

The order of the tables has been confirmed.

  1. Journal allows font size reduction (but no less than 8 pt. in size)

The fonts in the tables have been uniformly revised to 8 pt. in size.

  1. In some tables you can add horizontal lines

Horizontal lines have been added to the tables.

  1. Data in the tables should be better arranged, e.g. “Nonstandardized load factor” (Table 3)

The fonts in the tables have been revised to 8 pt. in size.

  1. The numbering of Tables A3-A6 described in the paragraph on lines 336-339 is unclear

Line346-347

The content has been revised to: The results are shown in Appendix A Tables A3 to A6.

  1. There are no appropriate table titles in subsection 3.2.2.

A new title has been added: New titles has been added to Tables A3 to A6

  1. Font unification (selected text, different fonts and font sizes appear, and tables appear in larger integers)

The fonts have been unified.

  1. Line 286, 352 – incomplete subchapter number

Line 286 is part of the content of section 3.2, and line 352 is part of the content of section 3.2.3. May I ask the reviewer to provide more detailed information on the areas that need to be revised? We will make the necessary revisions as soon as possible.

  1. The subsection should be written in italics, in accordance with the previous template: “Verification of the convergence validity” or “Suggestions for hospital volunteers”

Titles of subsections 3.2.1, 3.2.2, 3.2.3, 3.2.4, 4.2.1, 4.2.2, and 4.2.3 have been italicized.

  1. Authors' individual contributions must be provided

The authors' contributions are as follows:

Author Contributions

Conceptualization, writing – original draft, Kuan-Chieh Tseng;

Methodology, data curation, formal analysis, Investigation, Chun-Hao Yen;

Data curation, formal analysis, investigation, Chin-Shyang Shyu;

Formal analysis, methodology, Chih-Hung Tseng;

Supervision, writing – review & editing, Cheng-Ping Li;

Conceptualization, methodology, Fang-Wei Lin.

All authors have read and agreed to the published version of the manuscript

  1. The record of references should be corrected in accordance with the journal's requirements

The record of references has been revised in accordance with the journal’s requirements.

Round 2

Reviewer 3 Report (New Reviewer)

Comments and Suggestions for Authors

Thank you for your replies to the review. I don't have any more suggestions.

This manuscript is a resubmission of an earlier submission. The following is a list of the peer review reports and author responses from that submission.

Round 1

Reviewer 1 Report

Comments and Suggestions for Authors

Dear authors, thank you very much for the opportunity to read your paper. I find the topic interesting and well elaborated. However, I have one recommendation: the literature review is well done but does not reflect the discussion and interpretation of the results. I would recommend that the two parts be better linked. At the same time, I would recommend that a well-worded main message be included in the abstract.

Author Response

Reviewers’ Comments and Suggestions for Author (Round 1)

1 February 2024

Dear Reviewer,

Thank you for the constructive suggestions and comments on our manuscript(ID:sustainability-2764542). The suggestions and comments are helpful for improving the manuscript. We are submitting the revised version of the manuscript with our responses to the suggestions and comments by the reviewer. Many thanks for your guidance.

Our responses to each suggestion and comment are as follows, and they are presented in blue texts with a grey background color in the revised manuscript.

Dear authors, thank you very much for the opportunity to read your paper. I find the topic interesting and well elaborated. However, I have one recommendation: the literature review is well done but does not reflect the discussion and interpretation of the results. I would recommend that the two parts be better linked. At the same time, I would recommend that a well-worded main message be included in the abstract.

Response:

Thank you very much for your comments and suggestion. The modifications are as follows:

The findings of the study align with the literature review. For instance, in the discussion section, this paper asserts, “In terms of the relationships among the variables in this study, Hsieh (2014) focused on healthcare industry professionals, investigating the associations between resources, capabilities, and pharmaceutical employees’ well-being. The study revealed that job resources have a significant effect on subjective well-being. Therefore, the study hypothesis that volunteer work resources in hospitals significantly influence subjective well-being.” This passage echoes the literature review’s statement, “In terms of the relationships among the variables in this study, Hsieh (2014) focused on healthcare industry professionals, investigating the associations between resources, capabilities, and pharmaceutical employees’ well-being. The study revealed that job resources have a significant effect on subjective well-being. Therefore, the study hypothesis that volunteer work resources in hospitals significantly influence subjective well-being.”

Although the study subjects employed for hypothesis testing differ from healthcare volunteers or personnel, they can still be comparatively analyzed the results through the literature cited and the relationships among variables outlined in this paper.

Reviewer 2 Report

Comments and Suggestions for Authors

The article is of merit. However, the manuscript needs to be re-written.

1. The abstract could be more concise.

2. The placing of the citations is incorrect - the authors use the numeric citation without referring to the study, e.g. in the introduction text there is a sentence:

‘As [2] put it, if hospitals can effectively take advantage of volunteer manpower, the quality of medical services be ameliorated, and the social image of volunteer services can be better shaped’. 

This is not the correct way to use numerical citations and should be        written as follows: 

'If hospitals can effectively take advantage of volunteer manpower, the quality of medical services be enhanced and the social image of volunteer services better shaped [2]'.

This error is repeated several times throughout the manuscript.

3. In the research method section, Figure 1 has no citation. There is no sense of the measures used in the study. 

4. The presentation of the results is confusing to the reader. In addition, overall, there needs to be a tightening up of the academic writing. 

Comments on the Quality of English Language

There must be improvement in the quality of English and the way in which the authors refer to previous studies. 

Author Response

Reviewers’ Comments and Suggestions for Author (Round 1)

1 February 2024

Dear Reviewer,

Thank you for the constructive suggestions and comments on our manuscript(ID:sustainability-2764542). The suggestions and comments are helpful for improving the manuscript. We are submitting the revised version of the manuscript with our responses to the suggestions and comments by the reviewer. Many thanks for your guidance.

Our responses to each suggestion and comment are as follows, and they are presented in blue texts with a grey background color in the revised manuscript.

The article is of merit. However, the manuscript needs to be re-written.

  1. The abstract could be more concise.

Response:

Thank you very much for your comments and suggestion. The modifications are as follows:

Line22-34

Abstract: This study explored the subjective well-being of hospital volunteers during the COVID-19 pandemic from a mental health perspective. And adopted the purposive sampling method to conduct a questionnaire survey. A total of 520 questionnaires were distributed, and 500 questionnaires were recovered, with a recovery rate of 96.1%. The research hypotheses of this study were tested through SPSS 20.0 and AMOS 20.0 statistical analysis software. The results show that: (1) the work resources of the hospital volunteers exerted a significant impact on their subjective well-being; (2) the work resources of the hospital volunteers exerted no significant impact on their flow experience; (3) the work resources of the hospital volunteers exerted a significant impact on their leisure involvement; (4) the leisure involvement of the hospital volunteers exerted no significant impact on their subjective well-being; (5) the leisure involvement of the hospital volunteers exerted a significant impact on their flow experience; (6) the flow experience of the hospital volunteers exerted no significant impact on their subjective well-being; and (7) the leisure involvement of the hospital volunteers exerted a mediating effect between work resources and flow experience.

  1. The placing of the citations is incorrect - the authors use the numeric citation without referring to the study, e.g. in the introduction text there is a sentence:

‘As [2] put it, if hospitals can effectively take advantage of volunteer manpower, the quality of medical services be ameliorated, and the social image of volunteer services can be better shaped’. 

This is not the correct way to use numerical citations and should be        written as follows: 

'If hospitals can effectively take advantage of volunteer manpower, the quality of medical services be enhanced and the social image of volunteer services better shaped [2]'.

This error is repeated several times throughout the manuscript.

Response:

Thank you very much for your comments and suggestion. The modifications are as follows:

Line49

Line54

Line64

Line87

Line89

Line100

Line106

Line119

Line129

Line131

Line139

Line152

Line156

Line171

Line176

Line181

Line183

Line189

Line200

Line205

Line209

Line213

Line217

Line221

  1. In the research method section, Figure 1 has no citation. There is no sense of the measures used in the study. 

This paper has systematically organized the relationships among variables in the literature review. For instance, in examining the interplay of variables in the study, Hsieh (2014) focused on professionals within the pharmaceutical industry, investigating the associations between resources, capabilities, and the well-being of pharmaceutical employees. The study revealed that job resources significantly influence subjective well-being. Therefore, the present study hypothesized a significant impact of hospital volunteer work resources on subjective well-being. In other words, the research framework depicted in Figure 1 is constructed based on past relevant studies and pertinent literature, illustrating the interconnected relationships among variables.

  1. The presentation of the results is confusing to the reader. In addition, overall, there needs to be a tightening up of the academic writing. 

Response:

Thank you very much for your comments and suggestion. The modifications are as follows:

Line372-373

Line420

Line432-433

Line472

Line500

  1. Discussion

5.1. Empirical results of the research hypotheses

Figure 2. Model of the relationships among the work resources, leisure involvement, flow experience, and subjective well-being of hospital volunteers under the COVID-19 epidemic

Table 12. Empirical results of the research hypotheses

Hypothesis

Path relationship

Path value

Supported or not

1

Work resources have a significant impact on subjective well-being.

0.28

Supported

2

Work resources have a significant impact on flow experience.

-0.10

Not supported

3

Work resources have a significant impact on leisure involvement.

0.85

Supported

4

Leisure involvement has a significant impact on subjective well-being.

0.40

Not supported

5

Leisure involvement has a significant impact on flow experience.

0.98

Supported

6

Flow experience has a significant impact on subjective well-being.

0.11

Not supported

7

Leisure involvement has a mediating effect between work resources and flow experience.

Supported

          *p <.05

According to Figure 2 and Table 12, H1 was supported; that is, work resources had a significant impact on subjective well-being. This research result is the same as that of [27]. The possible reason is that the hospitals provided the subject volunteers with diverse work content and flexible service time; therefore, they could obtain a fuller life with more positive emotions.

H2 was not supported; in other words, work resources had a significant impact on flow experience. This research result is different from that of [28]. The possible reason is that under the COVID-19 epidemic, the subject volunteers considered that they could not excel in volunteer services as scheduled due to an increase in their psychological and physiological costs, thus reducing the flow experience.

H3 was supported, meaning work resources had a significant impact on leisure involvement. This research result is the same as that of [29]. The possible reason is that when the hospitals provided an environment in which the subject volunteers could achieve their work goals easier and reduce their physical and psychological costs associated with work, they were more willing to put effort into voluntary services.

H4 was not supported; in other words, leisure involvement had no significant impact on the flow experience. This research result is different from that of [42]. The possible reason might be that although the subject volunteers were engaged in voluntary services, the work content may have been too simple or monotonous, leading to an imbalance between challenges and abilities and a lack of flow experience.

H5 was supported, indicating that leisure involvement had a significant impact on subjective well-being. This research result is the same as that of [43]. The possible reason is that the subject volunteers were engaged in voluntary services. Under the concept of giving more than receiving, the volunteers were able to obtain more positive emotions. At the same time, because of the enrichment of their lives, the volunteers’ life satisfaction was high, and thus higher subjective well-being could be attained.

H6 was not supported; in other words, flow experience had no significant impact on subjective well-being. This research result is different from that of [7]. The possible factor is that the service content of the subject volunteers was too monotonous, and the subjects’ challenges and abilities were not balanced. Services were provided only to pass the time, resulting in the failure to produce higher subjective well-being.

H7 was supported, indicating leisure involvement had a mediating effect on work resources and flow experience. This research result is the same as that of [29]. The possible reason is that when the hospitals provided more support to the subject volunteers (such as epidemic prevention materials, flexible scheduling times, and positive incentives), the volunteers became more willing to participate in different challenges of voluntary services because of sufficient resources. Moreover, due to the high involvement of the hospital volunteers, their flow experience could be further affected.

5.2. Conclusion

This study took hospital volunteers as the research subjects and conducted an empirical study on the subjective well-being of hospital volunteers during the COVID-19 pandemic from a mental health perspective. From the research results, it was concluded that hospital volunteers’ work resources have a significant impact on subjective well-being and leisure involvement. The leisure involvement of hospital volunteers has a significant impact on flow experience and a mediating effect between work resources and flow experience. Work resources have no significant impact on flow experience, and leisure involvement and flow experience have no significant impact on subjective well-being. The results show that the research framework constructed according to the theory had some corresponding results after the empirical research.

5.3. Suggestions

5.3.1. Suggestions for hospital volunteers

The results of this study demonstrate that leisure involvement exerts a significant impact on flow experience. Therefore, it is suggested that hospital volunteers continue to engage in training courses to improve themselves. Especially with the progress of digital technology, more medical-related services are becoming electronic and paperless, such as Taiwan’s National Health Insurance app, E-health Pay app, and network registration system. Such training courses, in addition to reducing the digital gap, could allow volunteers to better assist patients or their families in completing relevant procedures and processes. For example, under the COVID-19 epidemic, in addition to Taiwan’s COVID-19 Vaccination Record Cards, the National Health Insurance Mobile APP also contains relevant content vaccination records. Familiarity with the operation of the National Health Insurance Mobile APP could enable volunteers to provide information to patients or their families in a timely manner when giving voluntary services. In the meanwhile, when hospitals need relevant volunteer manpower, after being equipped with relevant skills that are balanced with challenges, volunteers could have opportunities to produce a flow experience and show more enthusiasm when providing services to patients or their family members.

The results of this study reveal that flow experience had no significant impact on subjective well-being. Therefore, it is suggested that hospital volunteers not only improve themselves to provide better services and counseling but also seek emotional outlets for matters they have to face in the hospital every day, such as birth, aging, illness, and death, as well as the anxiety of patients and their family members. Such emotional outlets could help volunteers avoid emotional disorders that increase their psychological costs and cause emotional exhaustion. Therefore, it is suggested that hospital volunteers participate in social activities and better integrate into volunteer groups to benefit from having closer emotional connections with the hospital volunteer groups. Volunteers are also advised to enhance their positive emotions and mental health by sharing their experiences with other volunteers, which can lead to satisfaction and a richer life.

This study showed that leisure involvement had a mediating effect between work resources and flow experience. It is suggested that hospital volunteers continue to pursue the sense of pleasure and achievement brought by voluntary services based on the appropriate resources provided by the hospitals, such as sharing the service experience of special cases with volunteer groups. When these volunteers share their own experiences with new volunteers, voluntary services in hospitals can be promoted constantly. That is, they offer services to the served and gain self-realization from growth opportunities. Moreover, when hospital volunteers commit more and their professional abilities and task difficulties reach a balance, they will be better able to determine what kind of volunteer activities they want to do, and they will realize they have more opportunities to learn new things and make progress.

5.3.2. Suggestions for hospital administrators

The results of this study indicate that work resources had a significant impact on subjective well-being and leisure involvement. Therefore, it is suggested that hospital management units adjust or modify the service content, management system, and supervision of hospital volunteers in response to COVID-19. These adjustments can avoid poor work design or excessive work requirements that deplete the physical and psychological energies of hospital volunteers and cause them to be unable to obtain happiness from or become unwilling to participate in voluntary activities. Specifically, hospital management units can re-plan their volunteer manpower needs, as well as the service times and places, according to the retention and suspension of stations and the number of people on duty during the epidemic prevention period. For example, volunteer manpower can be arranged at the lobby service desk, the emergency and health inspection center, and other places. In addition, hospitals should also implement epidemic prevention and control measures, urge volunteers to perform disinfection and cleaning before and after providing services as well as wear masks, and require them to comply with epidemic prevention regulations to reduce the risk of infection. Furthermore, hospital administrators can continue to communicate with hospital volunteers. For example, hospitals are often tightly controlled against epidemic infection. In particular, admission staff must check health insurance cards, inquire about travel histories, and measure temperatures. Hospital administrators can also timely convey the idea that by volunteering in hospitals, volunteers can obtain more of the latest information about the COVID-19 epidemic and strengthen their sense of self-protection. In other words, through the support of hospital management units, hospital volunteers can provide appropriate resources (such as epidemic prevention materials), so that they can see that voluntary services are under acceptable control. In this way, hospital volunteers can trigger their positive psychological motivation, reduce their work pressure and emotional burden, and achieve happiness by providing voluntary services while promoting mental health.

5.3.3. Relevant research in the future

This study only provided self-reported questionnaires to volunteers. However, the organizational cultures and systems of hospitals can vary, such as the leadership models and incentive systems used for volunteer groups, which could not be explored in depth by this study. Besides, the emotional exhaustion faced by volunteers in different hospital units can also vary. For example, patients and their families in the health examination department and those at the lobby service desk have different characteristics. Limited by the length of this paper, this study did not include the above issues in the discussion. The inclusion of management system issues in future research on the psychological aspect of hospital volunteers could help to provide a more complete content.

There must be improvement in the quality of English and the way in which the authors refer to previous studies. 

Reviewer 3 Report

Comments and Suggestions for Authors

The work is relevant, but the results are presented very poorly.

More positive sides.

The relevance of the study is undoubted.

The literature review is significant.

The number of subjects is sufficient.

The survey procedure is well described.

Negative sides.

  There is a lack of understanding on the following issues.

What data was collected, what indicators were highlighted for further analysis. How exactly were they obtained? What scales were used to convert responses into scores? What methods of mathematical statistics were used?

Inaccuracy in the description of the study design raises many questions.

Let me give you a few examples.

Abstract.

. All the results in it are presented qualitatively and it is not clear how they were obtained. “4) the leisure involvement of the hospital volunteers exerted no significant impact on their subjective well-being.” How did the authors study the impact? This is not an experimental study. It is impossible to draw conclusions about the impact from the survey.

3.Research Method

The methods for collecting questionnaires are described well.

But!

1) It is not clear what indicators were obtained using these questionnaires. Recommendation: place the text of the questionnaire in the Appendix to the article.

2) It is not clear what indicators were taken for analysis. And how they were received. What scales were used for this? Recommendation: list all the indicators that were further used and describe how they were obtained, what their range of values is.

3) There is no description of the statistical methods used later in the article. Are they parametric or nonparametric statistics?

3.2. Research hypotheses

Incomprehensible design creates incomprehensible results. For example.

1) Tables with the results of factor analysis are presented. There are 4 tables in total.

I don’t understand, the authors conducted a factor analysis and they identified 4 factors? Then each table is a description of 1 factor? Or were there 4 different factor analyses?

2) There is no explanation after the tables.

For example, what is: Involvement 1, Involvement 4, Involvement 6? I didn’t find any explanation of this in the article at all.

A similar note applies to all tables.

I cannot assess the objectivity of the conclusions with such a presentation of the results.

Author Response

Reviewers’ Comments and Suggestions for Author (Round 1)

1 February 2024

Dear Reviewer,

Thank you for the constructive suggestions and comments on our manuscript(ID:sustainability-2764542). The suggestions and comments are helpful for improving the manuscript. We are submitting the revised version of the manuscript with our responses to the suggestions and comments by the reviewer. Many thanks for your guidance.

Our responses to each suggestion and comment are as follows, and they are presented in blue texts with a grey background color in the revised manuscript.

The work is relevant, but the results are presented very poorly.

More positive sides.

The relevance of the study is undoubted.

The literature review is significant.

The number of subjects is sufficient.

The survey procedure is well described.

Negative sides.

  There is a lack of understanding on the following issues.

What data was collected, what indicators were highlighted for further analysis. How exactly were they obtained? What scales were used to convert responses into scores? What methods of mathematical statistics were used?

The study employed a questionnaire survey method with hospital volunteers as the research subjects. Data collection involved the utilization of scales related to leisure involvement, flow experience, job resources, subjective well-being. The Likert five-point scale was utilized for the assessment, wherein respondents assign scores ranging from 1 to 5 based on their level of agreement, with response options spanning from "strongly disagree" to "strongly agree."

The study employed descriptive statistics using SPSS 20.0 statistical software to analyze demographic variables. Additionally, confirmatory factor analysis was conducted using AMOS 20.0 statistical software to validate the measurement model. Furthermore, a path model within the structural equation modeling framework was employed to analyze the causal relationships among variables.

Inaccuracy in the description of the study design raises many questions.

Let me give you a few examples.

Abstract.

. All the results in it are presented qualitatively and it is not clear how they were obtained. “4) the leisure involvement of the hospital volunteers exerted no significant impact on their subjective well-being.” How did the authors study the impact? This is not an experimental study. It is impossible to draw conclusions about the impact from the survey.

The study, based on the literature, proposed a research framework and formulated Hypothesis 4 (H4), which posits that "the leisure involvement of hospital volunteers exerts no significant impact on their subjective well-being." Subsequently, utilizing a Likert five-point scale, hospital volunteers, chosen as the research subjects, responded to measurements of leisure involvement and subjective well-being in the study. The path model within the structural equation modeling framework was then employed to analyze the causal relationship between leisure involvement and subjective well-being among hospital volunteers, validating the research hypothesis. Following path coefficient examination, with a value of 0.40, H4 was not supported, indicating that the leisure involvement of hospital volunteers does not significantly influence their subjective well-being.

3.Research Method

The methods for collecting questionnaires are described well.

But!

1) It is not clear what indicators were obtained using these questionnaires. Recommendation: place the text of the questionnaire in the Appendix to the article.

The questionnaire has been included in the appendix of the article.

Line511-515

Appendix A

Table A1. Questionnaire on Discussing the Subjective Well-being of Hospital Volunteers during the COVID-19 Pandemic from a Mental Health Perspective.

Dimension

Observed variables

Contents

Flow experience

Flow1

1. I have a clear goal of participating in hospital volunteer activities.

Flow2

2. I am clear about what I want to accomplish when I participate in hospital volunteer activities.

Flow3

3. I feel that participating in hospital volunteer activities is a worthwhile experience.

Flow4

4. I know what I want to do when I participate in hospital volunteer activities.

Flow5

5. I have a deep feeling about the experience of volunteering at the hospital.

Flow6

6. I would like to repeat the experience of volunteering at the hospital.

Flow7

7. I found the experience of participating in hospital volunteer activities enjoyable.

Flow8

8. I have a strong desire to participate in hospital volunteer activities.

Flow9

9. when I participate in hospital volunteer activities, my attention is focused on the activities.

Flow10

10. I have the ability to face challenges when volunteering in a hospital.

Flow11

11. I have the ability to perform the hospital volunteer activities that I am currently engaged in.

Flow12

12. when I participate in hospital volunteer activities, I feel that I am able to take on more difficult activities than I am doing now.

Flow13

13. I feel that my volunteer skills are getting better and better as I participate in hospital volunteer activities.

Flow14

14. I can fully demonstrate my abilities when participating in hospital volunteer activities.

Flow15

15. I feel in control when I am involved in hospital volunteer activities.

Flow16

16. I am in control of what I am responsible for when I participate in hospital volunteer activities.

Flow17

17. I don't worry about not doing well enough when I participate in hospital volunteer activities.

Flow18

18. I can remember everything that happens when I am involved in hospital volunteering.

Flow19

19. When I am involved in hospital volunteering, I do not have to deal with my current problems in life.

Flow20

20. Time passes without me realizing it when I am involved in hospital volunteering.

Flow21

21. My involvement in hospital volunteer activities is spontaneous.

Flow22

22. I volunteer in hospitals naturally and do not need to force myself.

Flow23

23. I know how to do things well when I am involved in hospital volunteer activities.

Flow24

24. When I am involved in hospital volunteer activities, I am able to carry out hospital volunteer activities spontaneously.

Leisure involvement

Involvement 1

1. Getting involved in hospital volunteer activities is important to me.

Involvement 2

2. Getting involved in hospital volunteer activities makes me feel fulfilled.

Involvement 3

3. Getting involved in hospital volunteer activities makes me feel happy.

Involvement 4

4. I am interested in participating in hospital volunteer activities.

Involvement 5

5. It is a pleasure for me to participate in hospital volunteer activities.

Involvement 6

6. I can show my true self when I engage in hospital volunteer activities.

Involvement 7

7. I can tell what kind of person someone is by the way he or she engages in hospital volunteer activities.

Involvement 8

8. Hospital volunteering has become a personal characteristic of mine and has formed an impression of me.

Involvement 9

9. I find that hospital volunteer activities are closely related to my life.

Involvement 10

10. I find that hospital volunteering plays an important role in my life.

Involvement 11

11. I find that most of my life is related to my hospital volunteer activities.

Work resources

Resource1

1. I have the opportunity to develop my strengths in hospital volunteer activities.

Resource2

2. I am able to fully express myself in hospital volunteer activities.

Resource3

3. I have the opportunity to learn new things when I volunteer in hospitals.

Resource4

4. I can decide by myself how I want to work in hospital volunteer activities.

Resource5

5. I can decide by myself the time I want to leave the venue of hospital volunteer activities.

Resource6

6. I can decide my own goals for my hospital volunteer activities.

Resource7

7. I can decide the order of my hospital volunteer activities.

Resource8

8. I can evaluate my own hospital volunteer activities.

Resource9

9. I can suspend my hospital volunteer activities.

Resource10

10. I can decide the amount of hospital volunteer work I need to accomplish within a certain period of time.

Resource11

11. I can decide by myself the pace of my hospital volunteer activities.

Resource12

12. I can decide by myself how long I want to do my hospital volunteering activities.

Resource13

13. I can decide by myself what kind of hospital volunteering activities I want to participate.

Subjective well-being

Well-being1

1. I feel that I am living my life in high spirits.

Well-being2

2. I feel that I am living a very happy life.

Well-being3

3. I feel very relaxed about my life.

Well-being4

4. I feel that my life is full of joy.

Well-being5

5. I feel confident.

Well-being6

6. I am living the lifestyle I want to live.

Well-being7

7. I am very satisfied with my current lifestyle.

Well-being8

8. I don't want to change my current life.

Well-being9

9. I feel in control of my life.

Well-being10

10. I enjoy life every day.

2) It is not clear what indicators were taken for analysis. And how they were received. What scales were used for this? Recommendation: list all the indicators that were further used and describe how they were obtained, what their range of values is.

The study utilized the AMOS software to conduct confirmatory factor analysis, and the assessment criteria are examined as follows:

1.Convergent validity assessment indicators:

(1) Standardized factor loadings should exceed 0.5.

(2) Composite reliability (C.R.) values should surpass 0.6.

(3) Average Variance Extracted (AVE) values should be greater than 0.5.

The aforementioned criteria are based on the convergent validity assessment standards proposed by Fornell and Larcker (1981).

2.Discriminant validity assessment indicators:

Correlation coefficients between constructs, when not encompassing the value of 1 within the 95% confidence interval, can be considered indicative of discriminant validity (Torkzadeh, Koufteros, & Pflughoeft, 2003).

3) There is no description of the statistical methods used later in the article. Are they parametric or nonparametric statistics?

The study employed parametric statistics, utilizing the SPSS 20.0 statistical software to conduct descriptive statistics on the sample of participants. The distribution of each item was analyzed through methods such as frequency distribution and percentages. Additionally, the structural equation model, specifically the path model within AMOS 20.0, was employed to analyze and validate the causal relationships among variables, in accordance with the research hypotheses.

3.2. Research hypotheses

Incomprehensible design creates incomprehensible results. For example.

1) Tables with the results of factor analysis are presented. There are 4 tables in total.

I don’t understand, the authors conducted a factor analysis and they identified 4 factors? Then each table is a description of 1 factor? Or were there 4 different factor analyses?

The study conducted a factor analysis that verified four factors, namely work resources, leisure involvement, flow experience, and subjective well-being. The results for each factor are presented in separate tables. Table 1 illustrates the work resources factor, Table 2 presents the leisure involvement factor, Table 3 displays the flow experience factor, and Table 4 outlines the subjective well-being factor.

2) There is no explanation after the tables.

I have responded within the main text.

Line290-295

This study tested the convergence validity of the dimensions including work resources. The factor loads of all dimensions were between 0.72 and 0.85; the composite reliability (CR) values were between 0.71 and 0.93; and the average variance extracted (AVE) values were between 0.55 and 0.68. All of these values conformed to the proposed convergence validity criteria [35-37], indicating this study had good convergence validity, as shown in Tables 2.

Line300-305

This study tested the convergence validity of the dimensions including Leisure involvement. The factor loads of all dimensions were between 0.69 and 0.93; the composite reliability (CR) values were between 0.75 and 0.91; and the average variance extracted (AVE) values were between 0.50 and 0.78. All of these values conformed to the proposed convergence validity criteria [35-37], indicating this study had good convergence validity, as shown in Tables 3.

Line310-315

This study tested the convergence validity of the dimensions including Flow experience. The factor loads of all dimensions were between 0.66 and 0.91; the composite reliability (CR) values were between 0.81 and 0.88; and the average variance extracted (AVE) values were between 0.53 and 0.66. All of these values conformed to the proposed convergence validity criteria [35-37], indicating this study had good convergence validity, as shown in Tables 4.

Line320-325

This study tested the convergence validity of the dimensions including Well-being. The factor loads of all dimensions were between 0.83 and 0.94; the composite reliability (CR) values were between 0.93 and 0.94; and the average variance extracted (AVE) values were between 0.77 and 0.83. All of these values conformed to the proposed convergence validity criteria [35-37], indicating this study had good convergence validity, as shown in Tables 5.

For example, what is: Involvement 1, Involvement 4, Involvement 6? I didn’t find any explanation of this in the article at all.

Line511-515

Appendix A

Table A1. Questionnaire on Discussing the Subjective Well-being of Hospital Volunteers during the COVID-19 Pandemic from a Mental Health Perspective.

Dimension

Observed variables

Contents

Flow experience

Flow1

1. I have a clear goal of participating in hospital volunteer activities.

Flow2

2. I am clear about what I want to accomplish when I participate in hospital volunteer activities.

Flow3

3. I feel that participating in hospital volunteer activities is a worthwhile experience.

Flow4

4. I know what I want to do when I participate in hospital volunteer activities.

Flow5

5. I have a deep feeling about the experience of volunteering at the hospital.

Flow6

6. I would like to repeat the experience of volunteering at the hospital.

Flow7

7. I found the experience of participating in hospital volunteer activities enjoyable.

Flow8

8. I have a strong desire to participate in hospital volunteer activities.

Flow9

9. when I participate in hospital volunteer activities, my attention is focused on the activities.

Flow10

10. I have the ability to face challenges when volunteering in a hospital.

Flow11

11. I have the ability to perform the hospital volunteer activities that I am currently engaged in.

Flow12

12. when I participate in hospital volunteer activities, I feel that I am able to take on more difficult activities than I am doing now.

Flow13

13. I feel that my volunteer skills are getting better and better as I participate in hospital volunteer activities.

Flow14

14. I can fully demonstrate my abilities when participating in hospital volunteer activities.

Flow15

15. I feel in control when I am involved in hospital volunteer activities.

Flow16

16. I am in control of what I am responsible for when I participate in hospital volunteer activities.

Flow17

17. I don't worry about not doing well enough when I participate in hospital volunteer activities.

Flow18

18. I can remember everything that happens when I am involved in hospital volunteering.

Flow19

19. When I am involved in hospital volunteering, I do not have to deal with my current problems in life.

Flow20

20. Time passes without me realizing it when I am involved in hospital volunteering.

Flow21

21. My involvement in hospital volunteer activities is spontaneous.

Flow22

22. I volunteer in hospitals naturally and do not need to force myself.

Flow23

23. I know how to do things well when I am involved in hospital volunteer activities.

Flow24

24. When I am involved in hospital volunteer activities, I am able to carry out hospital volunteer activities spontaneously.

Leisure involvement

Involvement 1

1. Getting involved in hospital volunteer activities is important to me.

Involvement 2

2. Getting involved in hospital volunteer activities makes me feel fulfilled.

Involvement 3

3. Getting involved in hospital volunteer activities makes me feel happy.

Involvement 4

4. I am interested in participating in hospital volunteer activities.

Involvement 5

5. It is a pleasure for me to participate in hospital volunteer activities.

Involvement 6

6. I can show my true self when I engage in hospital volunteer activities.

Involvement 7

7. I can tell what kind of person someone is by the way he or she engages in hospital volunteer activities.

Involvement 8

8. Hospital volunteering has become a personal characteristic of mine and has formed an impression of me.

Involvement 9

9. I find that hospital volunteer activities are closely related to my life.

Involvement 10

10. I find that hospital volunteering plays an important role in my life.

Involvement 11

11. I find that most of my life is related to my hospital volunteer activities.

Work resources

Resource1

1. I have the opportunity to develop my strengths in hospital volunteer activities.

Resource2

2. I am able to fully express myself in hospital volunteer activities.

Resource3

3. I have the opportunity to learn new things when I volunteer in hospitals.

Resource4

4. I can decide by myself how I want to work in hospital volunteer activities.

Resource5

5. I can decide by myself the time I want to leave the venue of hospital volunteer activities.

Resource6

6. I can decide my own goals for my hospital volunteer activities.

Resource7

7. I can decide the order of my hospital volunteer activities.

Resource8

8. I can evaluate my own hospital volunteer activities.

Resource9

9. I can suspend my hospital volunteer activities.

Resource10

10. I can decide the amount of hospital volunteer work I need to accomplish within a certain period of time.

Resource11

11. I can decide by myself the pace of my hospital volunteer activities.

Resource12

12. I can decide by myself how long I want to do my hospital volunteering activities.

Resource13

13. I can decide by myself what kind of hospital volunteering activities I want to participate.

Subjective well-being

Well-being1

1. I feel that I am living my life in high spirits.

Well-being2

2. I feel that I am living a very happy life.

Well-being3

3. I feel very relaxed about my life.

Well-being4

4. I feel that my life is full of joy.

Well-being5

5. I feel confident.

Well-being6

6. I am living the lifestyle I want to live.

Well-being7

7. I am very satisfied with my current lifestyle.

Well-being8

8. I don't want to change my current life.

Well-being9

9. I feel in control of my life.

Well-being10

10. I enjoy life every day.

A similar note applies to all tables.

I cannot assess the objectivity of the conclusions with such a presentation of the results.

Reviewer 4 Report

Comments and Suggestions for Authors

Tables and figures need re formatting. Integrate statistical results into text (rather than repetition of reporting that Hypotheses were supported without inclusion of results). Conclusions/implications can also be expanded for greater cohesion and impact. Suggest improving synthesis of existing literature and how current study contributes and build on this work.

Author Response

Reviewers’ Comments and Suggestions for Author (Round 1)

1 February 2024

Dear Reviewer,

Thank you for the constructive suggestions and comments on our manuscript(ID:sustainability-2764542). The suggestions and comments are helpful for improving the manuscript. We are submitting the revised version of the manuscript with our responses to the suggestions and comments by the reviewer. Many thanks for your guidance.

Our responses to each suggestion and comment are as follows, and they are presented in blue texts with a grey background color in the revised manuscript.

Tables and figures need re formatting. Integrate statistical results into text (rather than repetition of reporting that Hypotheses were supported without inclusion of results). Conclusions/implications can also be expanded for greater cohesion and impact. Suggest improving synthesis of existing literature and how current study contributes and build on this work.

Response:

Thank you very much for your comments and suggestion. The modifications are as follows:

Previously, studies on hospital volunteers have predominantly approached their motivation and participation demands from the perspective of serious leisure (Deery, Jago & Mair, 2011; Cantillon & Baker, 2019), often overlooking the inclusion of job resources within the volunteer service unit. However, in the context of hospital volunteer services, factors such as interpersonal relationships with supervisors, support from colleagues, and team atmosphere can potentially influence the level of engagement of hospital volunteers in their voluntary service. Therefore, this paper introduces job resources as a research variable, not only expanding the scope of volunteer-related research but also contributing to practical applications in volunteer management.

On the other hand, existing literature reveals that relevant studies have mostly concentrated on managerial aspects (Aragaki, Saito, Takahashi & Kai, 2007; Hotchkiss, Fottler, Unruh, 2009; Doherty & Hoye, 2011) and psychological dimensions (Wymer, 199; Faulkner & Davies; Ferreira, Proença, & Proença, 2012). However, there is limited discourse on how hospital volunteers should address physiological and psychological issues they may face. This paper not only incorporates job resources into the discussion from a managerial standpoint but also employs the concept of leisure involvement as one of the strategies for addressing the physiological and psychological issues faced by hospital volunteers. Through empirical research in this paper, it is anticipated to provide guidance for recruitment and training within hospital volunteer management units, offering practical solutions to the challenges currently encountered by hospital volunteers.

Deery M., Jago L., Mair J. (2011) ‘Volunteering for Museums: The Variation in Motives across Volunteer Age Groups’, Curator: The Museum Journal 54(3): 313–25.

Cantillon Z., Baker S. (2019) ‘Serious Leisure and the DIY Approach to Heritage: Considering the Costs of Career Volunteering in Community Archives and Museums’, Leisure Studies, URL (consulted 2 March 2020): https://doi.org/10.1080/02614367.2019.1694571.

Hotchkiss RB, Fottler MD, Unruh L. Valuing volunteers. The impact of volunteerism on hospital performance. Health Care Manage Rev 2009; 34: 119–128.

Aragaki. M., Saito. T., Takahashi. M., & Kai. I. Hospital volunteer’s role and accident-prevention systems: a nationwide survey of Japanese hospitals. Health Serv Manage Res 2007; 20: 220–226. https://doi.org/10.1258/095148407782219021

 Ferreira, M.R., Proença, T., Proença, J.F. Motivation among hospital volunteers: an empirical analysis in Portugal. Int Rev Public Nonprofit Mark 2012; 9: 137–152.

Wymer, W,W. Hospital volunteers as customers: understanding their motives, how they differ from other volunteers, and correlates of volunteer intensity. J Nonprofit Public Sect Mark 1999; 6: 51–76.

Doherty A, Hoye R. Role ambiguity and volunteer board member performance in nonprofit sport organizations. Nonprofit Manag Leadersh 2011; 22: 107–128.

Faulkner M, Davies S. Social support in the healthcare setting: the role of volunteers. Health Soc Care Community 2005; 13: 38–45.
